# A Closer Look into Automatic Evaluation Using Large Language Models

**Cheng-Han Chiang**
National Taiwan University,
Taiwan
dcml0714@gmail.com

**Hung-yi Lee**
National Taiwan University,
Taiwan
hungyilee@ntu.edu.tw

## Abstract

Using large language models (LLMs) to evaluate text quality has recently gained popularity. Some prior works explore the idea of using LLMs for evaluation, while they differ in some details of the evaluation process. In this paper, we analyze *LLM evaluation* (Chiang and Lee, 2023)[1] and *G-Eval* (Liu et al., 2023), and we discuss how those details in the evaluation process change how well the ratings given by LLMs correlate with human ratings. We find that the auto Chain-of-Thought (CoT) used in G-Eval does not always make G-Eval more aligned with human ratings. We also show that forcing the LLM to output only a numeric rating, as in G-Eval, is suboptimal. Last, we reveal that asking the LLM to explain its own ratings consistently improves the correlation between the ChatGPT and human ratings and pushes state-of-the-art (SoTA) correlations on two meta-evaluation datasets.

## 1 Introduction

Large language models (LLMs) trained with task instructions and human feedback can follow natural language instructions to complete a task (Askell et al., 2021; Sanh et al., 2022; Wei et al., 2022a; Ouyang et al., 2022). Recently, the instruction-following ability of LLMs makes them promising candidates for automatic evaluation (Chiang and Lee, 2023; Liu et al., 2023; Wang et al., 2023; Huang et al., 2023). By simply instructing the LLMs on how to rate and giving the LLMs the sample to be rated, the LLM can follow the instructions and provide a rating of the sample.

Chiang and Lee (2023) propose *LLM evaluation* and Liu et al. (2023) propose *G-Eval*; both of which use LLMs to evaluate samples by giving the LLM instructions, and they both show that some LLMs can yield evaluation results that are aligned to the

---

[1] In this paper, the term *LLM evaluation* is used to refer to the specific method proposed by Chiang and Lee (2023).

evaluation results of humans. Still, LLM evaluation and G-Eval differ in some specific design choices in the evaluation procedure. Since Chiang and Lee (2023) and Liu et al. (2023) use distinct tasks, it is hard to know how the differences between LLM evaluation and G-Eval affect the evaluation results. This makes practitioners in the future hard to determine how to conduct an automatic evaluation using LLMs.

Given that LLM evaluation and G-Eval have already received significant attention shortly after publication, these methods will likely revolutionize the evaluation in NLP. Therefore, conducting a detailed analysis of these approaches is essential and timely. This paper aims to identify the crucial components in LLM evaluation and G-Eval that contribute to stronger correlations with human ratings. Based on our analysis, we provide guidelines on how to use LLMs for automatic evaluations. We have the following findings:

- Auto-CoT (proposed by G-Eval) does not always improve the correlation between LLM and human ratings.

- Making the LLMs output only a single numeric rating is suboptimal.

- Asking the LLMs to rationalize their own ratings significantly improves the correlation between the LLMs' ratings and human ratings.

- On two datasets, we improve the best correlation that ChatGPT's rating can achieve, and some correlations even exceed prior SoTA correlations obtained using the ratings of GPT-4 in Liu et al. (2023).

## 2 Experiment Setup

Our paper studies what components in LLM evaluation and G-Eval make the ratings generated by LLM correlate with human ratings better, and we aim to improve the correlation.

## 2.1 LLM as an Automatic Evaluation Metric

Both LLM evaluation (Chiang and Lee, 2023) and G-Eval (Liu et al., 2023) propose to ask LLMs to rate a sample regarding some attributes of the sample (e.g., fluency, grammaticality) using a $k$-point Likert scale. They give the LLMs (1) **descriptions of the rating task**, (2) the **definition and rating criteria** of the attribute to be rated, (3) the **sample to be rated**, and (4) **a sentence that prompts the LLM to give the rating**[2]. The LLM outputs a sequence containing the rating. Unless specified, we follow prior works to sample $N = 20$ sequences from the LLM and average those ratings as the final rating. While the two methods share the core concept, they differ in two details.

**Difference 1: Auto Chain-of-Thought** The task descriptions and rating criteria in LLM evaluation and G-Eval are all human-written. However, Liu et al. (2023) argue that some evaluated attributes require more than simple definition and evaluation criteria, so they use LLMs to determine the evaluation steps. Specifically, they concatenate the task description, definition, and criteria of the attributes and append a line "Evaluation steps:" to prompt the LLM. The LLM then generates an ordered list containing the step-by-step evaluation steps. They dub this process *auto chain-of-thought (CoT)*. G-Eval uses human-written task instructions and auto-CoT-generated evaluation steps to prompt the LLM to rate the sample.

**Difference 2: Prompts for Output** At the end of the input to LLMs, G-Eval uses the prompt "{{placeholder}} (score only):" to restrict the LLM to output **only the numeric rating**; the placeholder will be replaced by the evaluated attributes. In contrast, LLM evaluation uses the following question to ask the LLM to assign the rating: "How {{placeholder}} is the sample? (on a scale of 1-k, with 1 being the lowest)". The LLM's **output form is not restricted**.

## 2.2 Meta-Evaluating an Evaluation Metric

Given a sample, an evaluation metric assigns it a rating. To evaluate an evaluation metric, we need a dataset containing human ratings for samples in the dataset. We calculate the correlation coefficient between the ratings obtained by the evaluation metric and the human ratings. A higher correlation

---

[2]In our paper, we use different highlight colors to represent different parts of the prompt, as shown in the above text. Additionally, we use cyan to represent the parts generated by **auto Chain-of-Thought**

indicates the evaluation metric better aligns with human ratings. We adopt Pearson $r$ and Kendall's $\tau$ as they are widely used in meta-evaluations (Graham et al., 2015; Bojar et al., 2017; Zhang* et al., 2020). **In our paper, all the *correlation* refers to the correlation coefficient between the ratings of LLM and human ratings.** Details on the calculation of correlation coefficients are in Appendix C.

We use **SummEval** (Fabbri et al., 2021) and **Topical-Chat** (Gopalakrishnan et al., 2019; Mehri and Eskenazi, 2020) as the meta-evaluation datasets, following Liu et al. (2023). SummEval is a meta-evaluation dataset for summarization derived from the CNN/DailyMail dataset (Hermann et al., 2015). Each summary in SummEval is rated by humans based on the *coherence*, *consistency*, *fluency* of the summary, and *relevance* between the summary and the source document. Topical-Chat is a dataset that evaluates the quality of a response given the dialogue history and a piece of knowledge relating to the dialogue. We follow Zhong et al. (2022) to evaluate the *naturalness*, *coherence*, *engagingness*, and *groundedness* (whether the response is grounded on the provided knowledge) of the response. The dataset details are in Appendix E.

## 2.3 Large Language Models

An LLM used as an evaluation metric should be affordable and accessible to whoever wants to use it. Based on this principle, we use ChatGPT (gpt3.5-turbo-0613) (OpenAI, 2022) for evaluation since it has lower cost and improved performance compared with other GPT-3.5 models. ChatGPT is also used in LLM evaluation and G-Eval. While Liu et al. (2023) further use GPT-4 (OpenAI, 2023) in their experiments, we cannot use GPT-4 in our experiments since most people, including us, have limited or no access to GPT-4, making it utterly unsuitable as an evaluation metric.

In our preliminary experiments, we also try to use the best open LLM (at the time of writing this manuscript) on Open LLM leaderboard, the falcon-40b-instruct model (Almazrouei et al., 2023), but we find it cannot follow the instructions and rate the samples very well. Hence, we exclude open LLMs in our paper.

## 3 Better Usage of LLM for Evaluation

### 3.1 Is Auto CoT Always Useful?

Liu et al. (2023) shows that adding the evaluation steps generated by auto CoT improves the correla-

| Sec. | Ablations | | Coherence | | Consistency | | Fluency | | Relevance | |
|------|-----------|--------|-----------|--------|-------------|--------|---------|--------|-----------|--------|
| | CoT | Output | $r$ | $\tau$ | $r$ | $\tau$ | $r$ | $\tau$ | $r$ | $\tau$ |
| GPT-4[†] | ?[‡] | *Score only* | 0.581 | 0.463 | 0.575 | 0.419 | 0.6 | 0.457 | 0.599 | 0.409 |
| 3.1 | ✓ | *Score only* | 0.45 | 0.359 | 0.37 | 0.286 | 0.319 | 0.203 | 0.403 | 0.327 |
| | ✗ | | 0.344 | 0.248 | 0.328 | 0.185 | **0.361** | 0.177 | 0.353 | 0.248 |
| 3.2 | ✗ | *Score only* | 0.344 | 0.248 | 0.328 | 0.185 | **0.361** | 0.177 | 0.353 | 0.248 |
| | ✗ | *Free Text* | **0.46** | 0.342 | **0.476** | 0.334 | **0.477** | 0.273 | 0.324 | 0.228 |
| | ✗ | *Rate-explain* | **0.557** | 0.44 | **0.473** | 0.337 | **0.451** | 0.306 | **0.509** | 0.348 |
| | ✗ | *Analyze-rate* | **0.635** | 0.476 | **0.537** | 0.34 | **0.479** | 0.302 | **0.444** | 0.305 |

Table 1: The Pearson's $r$ and Kendall's $\tau$ correlation coefficient between LLMs' ratings and human ratings for SummEval. All the results in this table, except the first row, are from ChatGPT. We consider *auto CoT + score only* using ChatGPT proposed in G-Eval as the baseline of this paper. We **boldface** the Pearson's $r$ statistically significantly higher than the baseline (except GPT-4). †: results from Liu et al. (2023). Some numbers are different because we re-calculate the correlations based on the GPT-4 responses Liu et al. (2023) released. ‡: The results of GPT-4 cannot serve as a reasonable comparison since we find something odd in the prompts Liu et al. (2023) use, which we elaborate in Appendix A.

tion on SummEval when using GPT-4 for evaluation. By scrutinizing their results, we find that the correlations when using auto CoT and not using it often differ by less than 0.02. This raises two questions: (1) Is this difference statistically significant? (2) Does auto CoT yield higher correlations for different LLMs and datasets? To answer these questions, we use ChatGPT to rate the samples in SummEval and Topical-Chat using two sets of prompts, one with the evaluation steps generated using auto CoT and one without those evaluation steps. In this experiment, we follow G-Eval and restrict ChatGPT to output only a numeric score. Following Graham and Baldwin (2014), we use William's test for significance to see if the Pearson's $r$ of using and not using auto CoT is statistically significantly different. We try to follow the prompts used in G-Eval when possible; still, we have to construct some prompts since Liu et al. (2023) only release part of the prompts and some of which are problematic. We list all the prompts and how they are obtained in Appendix F.

The experiment results for SummEval are shown in the block in blue in Table 1. We also list the best results of G-Eval using GPT-4 from Liu et al. (2023) in the first row of Table 1 only for reference. Comparing our results with GPT-4 is unfair since we use ChatGPT, which is weaker than GPT-4. **A more reasonable baseline for our paper is the "*auto CoT + score only*" using ChatGPT on the second row**, which is the method proposed by G-Eval and shows the highest correlation that ChatGPT can achieve in Liu et al. (2023). The numbers here differ from results in Liu et al. (2023) because

we carefully reproduce their results ourselves.

Back to Table 1, we can see that auto CoT leads to higher correlations for *coherence*, *consistency*, and *relevance*. By William's test, these higher correlations reach statistical significance with $p$-values less than 0.05. However, using auto CoT results in a lower Pearson's $r$ for *fluency*, and this inferiority in Pearson's $r$ is also statistically significant.

The results for Topical-Chat are illustrated in Table 2. For Topical-Chat, the Pearson's $r$ of using and not using auto CoT are very close for all four attributes except *groundedness*, with differences less than 0.025, and these differences are not statistically significant. For *groundedness*, auto CoT even drastically decreases the correlation. In summary, using auto CoT does not yield consistent and meaningful improvements compared with not using CoT. This should not be surprising since the evaluation steps generated with auto CoT often merely paraphrases the evaluation criterion and instructions given to the LLM.

## 3.2 Prompt for Outputs

In this section, we explore if the difference in how ChatGPT is prompted to output makes it's ratings better aligned with human ratings. We use two sets of prompts that share the same task descriptions and evaluation criteria but differ in how they prompt the LLM to generate the output. One uses "score only", as in G-Eval. The other replaces the "score only" with "How {{placeholder}} is the sample? (on a scale of 1-k, with 1 being the lowest)", as in LLM evaluation. We call the latter prompts *free text* since they do not

| Sec. | Ablations | | Naturalness | | Coherence | | Engagingness | | Groundedness | |
|------|-----|--------|-----------|-----------|-----------|-----------|--------------|-----------|--------------|-----------|
| | CoT | Output | $r$ | $\tau$ | $r$ | $\tau$ | $r$ | $\tau$ | $r$ | $\tau$ |
| 3.1 | ✓ | *Score only* | 0.393 | 0.358 | 0.468 | 0.391 | 0.549 | 0.513 | 0.311 | 0.566 |
| | ✗ | | 0.408 | 0.331 | 0.443 | 0.404 | 0.557 | 0.535 | 0.358 | 0.582 |
| 3.2 | ✗ | *Score only* | 0.408 | 0.331 | 0.443 | 0.404 | 0.557 | 0.535 | **0.358** | 0.582 |
| | ✗ | *Free Text* | **0.464** | 0.476 | 0.524 | 0.426 | **0.611** | 0.557 | **0.563** | 0.666 |
| | ✗ | *Rate-explain* | **0.524** | 0.47 | 0.477 | 0.416 | 0.567 | 0.524 | **0.58** | 0.693 |
| | ✗ | *Analyze-rate* | **0.573** | 0.47 | 0.486 | 0.416 | **0.628** | 0.524 | **0.725** | 0.693 |

Table 2: The Pearson's $r$ and Kendall's $\tau$ correlation coefficient between LLMs' ratings and human ratings for Topical-Chat. All the results in this table, except the first row, are from ChatGPT. We **boldface** the Pearson's $r$ statistically significantly higher than *auto CoT + score only*. We underline the Pearson's $r$ comparable *auto CoT + score only*.

restrict the output form.

The results for SummEval are shown in the yellow blocks in Table 1, and the results for Topical-Chat are shown in Table 2. We find that allowing ChatGPT to respond to the question freely yields Pearson's $r$ and Kendall's $\tau$ much higher than restricting the model to output a single numeric score for almost all attributes of both datasets. The higher Pearson's $r$ of *free text* compared with *score only* is statistically significant. The only exception is the *relevance* of SummEval, where *free text* yields slightly lower correlations.

Initially, we thought ChatGPT aligns better with human ratings in *free text* because it can generate natural language explanations to justify their rating, making the ratings more correlated with human ratings. However, we observe that the responses of ChatGPT when prompted with *free text* mostly contain a single numeric rating, which is the same behavior when it is instructed by *score only*. This means that what the model is *allowed to generate* is more important than what it *really generates*.

The above observations make us curious if the correlations can be higher if ChatGPT is instructed to justify its ratings. Inspired by chain-of-thought in Wei et al. (2022b) and Kojima et al. (2022) (not the auto CoT in G-Eval), we ask ChatGPT to provide their reasoning and rationales on the ratings. Instead of asking ChatGPT to output only a score, we construct two types of prompts that ask ChatGPT to rationalize its decision. The first type of prompt, called *analyze-rate*, asks ChatGPT to analyze the samples regarding the evaluated criteria first and give the rating. The second type of prompt, called *rate-explain*, asks ChatGPT to provide the numeric ratings first and explain why it gives such a rating. *analyze-rate* is more like the zero-shot

chain-of-thought (Kojima et al., 2022). Refer to Appendix F.1.1 for the exact prompts we use.

The results of asking ChatGPT to explain/analyze how they rate the sample are shown in the last two rows in Table 1 and Appendix Table 2. We find that for all attributes of both datasets, *rate-explain* and *anlyze-rate* both lead to correlations stronger than or at least comparable to the correlation of asking ChatGPT to output only a numeric rating (*score only*). By asking ChatGPT to explain/analyze, we improve the best correlations that can be achieved by ChatGPT in Liu et al. (2023) (the *Auto-CoT + score only*). Moreover, when asked to explain/analyze when rating, ChatGPT's correlation can be better than or comparable to the state-of-the-art correlation coefficients obtained from GPT-4 in Liu et al. (2023) for *coherence* of SummEval and three attributes of Topical-Chat. We hypothesize that some attributes (e.g., *coherence* for SummEval) are harder for ChatGPT to rate, so the correlations for these attributes show a larger improvement when ChatGPT explains how it rates the sample.

In *rate-explain*, the output of ChatGPT contains a numeric rating followed by some explanations. As an auto-regressive language model, ChatGPT cannot depend on the explanation when generating the rating due to causal attention. If we stop the generation after ChatGPT generates the ratings, the output of *rate-explain* will only contain the ratings, just like the output forms in *score only*. Although the ratings in *rate-explain* do not depend on ChatGPT's rationales for the ratings, the ratings still correlate better with human ratings, compared with the ratings in *score only*. We think this is because when ChatGPT knows it needs to explain the ratings, it tends to generate ratings that are easier for it to explain, and a rating that is more

aligned to humans' rating is easier for ChatGPT to explain.

### 3.3 Empirical Guidelines

Based on the analysis and results in this section, we provide the following guideline: **Always ask ChatGPT to explain/analyze when rating.** We do not see *rate-explain* to be significantly better (or worse) than *analyze-rate*, so it is hard to determine which one to use. A valid method is sampling some ratings using *rate-explain* and sampling some ratings using *analyze-rate* and averaging the ratings from the two prompts as the final rating. Using auto CoT is optional since it does not always lead to higher correlations with human ratings. We also find that using auto CoT does not always improve the correlations when ChatGPT is asked to explain; this result is shown in Appendix Table 3.

### 3.4 Robustness of the Guidelines

LLMs are notorious for their performance fluctuation due to the input prompts, and the sequence generated by LLMs can be different when changing the hyperparameters used in decoding. To verify the validity of our empirical guidelines, we conduct the following two sets of experiments: (1) we vary the temperature used in sampling the output from ChatGPT, and (2) we vary the prompt given to ChatGPT.

#### 3.4.1 Varying the Temperature

We check if our guideline holds if we change the temperature $T$ during generation. We compare Pearson's $r$ when using the method proposed in G-Eval (Auto-CoT + score only) with *rate-explain* and *analyze-rate* under different temperatures used when generating the output from ChatGPT. We follow Chiang and Lee (2023) and use two temperatures: 0.7 and 0.3.

The results are shown in Appendix Table 5 and summarized as follows: First, when fixing the sampling temperature, we find that *rate-explain* and *analyze-rate* always achieve a higher correlation compared with G-Eval. This supports our guideline that "*asking the LLM to explain/analyze outperforms the method proposed in G-Eval.*" Next, we observe that the correlation of G-Eval when $T = 0.3$ is much lower than that of $T = 1.0$. This shows that G-Eval is not robust to sampling temperature. Contrarily, we find that the correlations obtained by *rate-explain* and *analyze-rate* do not significantly change for different sampling

temperatures for almost all cases. This shows that *rate-explain* and *analyze-rate* are more robust than G-Eval with respect to the sampling temperature.

#### 3.4.2 Changing the Prompts

We check if our guideline holds if we change the prompt given to ChatGPT. In this experiment, we changed the prompts to ChatGPT by appending some instructions before the descriptions of the rating task. We tried with two prompts: (1) the HHH prompts and (2) the human annotator prompts. The HHH prompt is designed by Bai et al. (2022) to align the output of LLMs to be more harmless, honest, and helpful. The human annotator prompt is inspired by Chiang and Lee (2023), who use a similar prompt to make the LLM behave as a human annotator. These two prompts will be inserted before the prompt we originally used in our paper. We use these two prompts to inject persona into the LLM. This is inspired by Zeng et al. (2023), which shows that the output of GPT3 can be different when prompted with a different persona. The prompts are detailed in Appendix F.3.

The results are shown in Table 6 and summarized as follows: *rate-explain* and *analyze-rate* consistently outperform the G-eval when using the human annotator prompts and the HHH prompts. This indicates that our guidelines are robust toward different prompts. We also find that the correlations of G-Eval significantly drop when adding the human-annotator prompts or HHH prompts. On the other hand, the correlation for *rate-explain* and *analyze-rate* do not significantly decrease when adding the human-annotator prompt and the HHH prompt. This shows that asking the LLM to explain is more robust to the variation of the prompts.

## 4 Conclusion

We study how to better use ChatGPT as an automatic evaluation tool by scrutinizing LLM evaluation and G-Eval. We provide concrete guidelines and show that by using those guidelines, the correlations of several evaluated attributes given by ChatGPT, a publicly usable model, can be higher than or comparable to the ratings given by GPT-4, a highly restricted and pricey model. We also show that the evaluation results based on our guidelines improve the best correlation that ChatGPT's rating can achieve. We believe our results and guidelines help future researchers better use LLMs for evaluation.

## Limitations

There are three main limitations of this paper.

1. We only use ChatGPT to conduct the experiments in this paper. We explain why we chose ChatGPT in Section 2.3. We believe that using ChatGPT is already enough since we show that the correlations obtained by using ChatGPT are already comparable to or better than the previous SoTA results obtained by GPT-4.

2. We only conduct analysis using two tasks, while we know that NLP has more diverse tasks. We do not guarantee that our observations can generalize to all the other datasets. We recommend the users verify the effectiveness of using LLM to evaluate the tasks of interest.

3. We cannot fairly compare our results with Liu et al. (2023), the previous SoTA results, due to multiple reasons. We explain those reasons in Appendix A.

## Ethics Statement

Our paper follows the ACL Code of Ethics. We do not see a particular harmful outcome of our paper. The code and datasets for reproducing our experiments can be found at `https://github.com/d223302/A-Closer-Look-To-LLM-Evaluation/`.

## Acknowledgements

We want to thank the reviews for providing detailed feedback and actionable suggestions, which helped us strengthen our paper. We also want to thank the senior committee members for monitoring the reviewing process. Cheng-Han Chiang is supported by a Ph.D. scholarship program by Delta Electronics.

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

## A Why We Cannot Fairly Compare with the Results in Liu et al. (2023)

As a work highly related to G-Eval, we would really like to compare our results with G-Eval. However, we encounter difficulties when comparing our results with those in Liu et al. (2023) for the following reasons.

- G-Eval proposes to use GPT-4 as the evaluation tool, while it is currently a highly restricted model, and we only have limited access to it.

- G-Eval only releases the prompts for SummEval. We need to construct the prompts for Topical-Chat based on the human evaluation instructions released by Mehri and Eskenazi (2020). It is possible that the prompts we use for Topical-Chat are different from the prompts used in Liu et al. (2023), making their results incomparable to ours.

- The prompts of *fluency* in SummEval released by Liu et al. (2023) in here is problematic so we need to construct new prompts for *fluency*. Refer to Appendix F.1 for detailed explanations. This makes us unable to directly compare our results with the results in Liu et al. (2023).

- We cannot reproduce the numbers on the paper of G-Eval even when using their official implementation and the GPT-4 responses they release. This means that the only thing we

do is calculate the correlation coefficient using the data and code released on the official GitHub of G-Eval, but the numbers are quite different from the results in G-Eval's paper. Moreover, the results of *fluency* they provide is the result not using auto CoT, but the results of the other three attributes for SummEval use auto CoT. That is why we use a question mark for the auto CoT field in Table 1.

- The Table 2 in Liu et al. (2023) seems to be wrong. The caption (Spearman's $\rho$ and Kendall's $\tau$) does not match the headers ($r$ and $\rho$). This makes us hard to compare their results with ours reliably.

## B  Supplementary Results for Topical-Chat

Table 2 is the supplementary results for Topical-Chat that we referred to in the main content. We plan to move Table 2 to the main content using the additional one page in the camera-ready version if the paper is accepted. See how Pearson's $r$ and Kendall's $\tau$ are calculated in Appendix C.

### B.1  Is Auto CoT Useful When ChatGPT Is Asked to Explain?

In Table 3, we show the results when we add the evaluation steps generated by auto CoT when we ask ChatGPT when prompting with (*rate-explain*). We find that on *groundedness*, using auto CoT is worse. However, for the other three attributes, using auto CoT is better. This again shows that auto CoT is not particularly useful.

## C  Calculation of Correlation Coefficient

In this paper, we calculate Pearson's $r$ and Kendall's $\tau$ between human ratings and ChatGPT's ratings. Whether to use Spearman's rank correlation or Pearson's (linear) correlation to evaluate the alignment between human ratings and an automatic evaluation metric is long-standing, but there has been an increasing trend towards Pearson's correlation since 2014 (Macháček and Bojar, 2014; Graham and Baldwin, 2014; Zhang* et al., 2020). We use the `pearsonr` and `kendalltau` in `scipy.stats` for calculating the correlation coefficients. For each attribute of each sample, the rating of ChatGPT is obtained by 20 samples; we set the decoding temperature to 1 and the top-$p$ in nucleus sampling to 1, following G-Eval (Liu et al., 2023).

Consider a dataset with $N$ source documents, and each source document has $M$ corresponding target documents. We also have the human ratings for $N \cdot M$ target documents on a specific attribute. While each attribute of each target document is rated by more than one human rater, we average those ratings when calculating the correlation coefficient. So the $N \cdot M$ ratings are the average ratings from different raters. In the case of SummEval, we have $N = 100$ source documents and $M = 16$ summaries generated by 16 summarization models. There are two different methods for calculating correlation coefficients.

### C.0.1  Method 1: Dataset-Level Correlation Coefficient

In this method, we first obtain the ratings on $N \cdot M$ target documents from ChatGPT. We then calculate the correlation coefficient between the $N \cdot M$ ChatGPT's ratings and the $N \cdot M$ average human ratings. In this case, the correlation coefficient is calculated among two $N \cdot M$ vectors, meaning that the correlation coefficient is calculated across the entire dataset.

### C.0.2  Method 2: Document-Level Correlation Coefficient

In this method, for each source document, we obtain the ratings of its $M$ target documents using ChatGPT. Next, we calculate the correlation coefficient between these $M$ ChatGPT ratings and the corresponding $M$ human ratings. After iterating the above process over all the $N$ source documents, we obtain the $N$ correlation coefficients. We average the $N$ correlation coefficients as the final correlation coefficient. In this case, the correlation coefficient is calculated at the document-level and averaged over the whole dataset.

### C.1  How We Calculate the Correlation Coefficient

In Table 1 and 2 in this paper, we use Method 1 (Subsection C.0.1) to calculate Pearson's correlation, following the recommendation in Graham et al. (2015). Calculating the correlation coefficient on the dataset level is also used in LLM evaluation (Chiang and Lee, 2023). **Calculating a single correlation coefficient on the dataset level allows us to use William's test** to test whether two Pearson's $r$ are significantly different.

For Kendall's $\tau$ in Table 1 and 2, we follow most prior works (Zhong et al., 2022; Liu et al., 2023) to

| Sec. | Ablations | | Naturalness | | Coherence | | Engagingness | | Groundedness | |
|---|---|---|---|---|---|---|---|---|---|---|
| | CoT | Output | $r$ | $\tau$ | $r$ | $\tau$ | $r$ | $\tau$ | $r$ | $\tau$ |
| | ✗ | *Score only* | 0.393 | 0.358 | 0.468 | 0.391 | 0.549 | 0.513 | 0.311 | 0.566 |
| 3.2 | ✓ | *rate-explain* | **0.554** | 0.478 | **0.512** | 0.429 | **0.613** | 0.566 | **0.555** | 0.664 |
| | ✗ | *rate-explain* | **0.524** | 0.47 | 0.477 | 0.416 | 0.567 | 0.524 | **0.58** | 0.693 |

Table 3: The Pearson's $r$ and Kendall's $\tau$ correlation coefficient between LLMs' ratings and human ratings for Topical-Chat. All the results in this table, except the first row, are from ChatGPT. We **boldface** the Pearson's $r$ statistically significantly higher than *auto CoT + score only*. We underline the Pearson's $r$ comparable *auto CoT + score only*.

calculate Kendall's $\tau$ using Method 2 (document-level, Section C.0.2) to understand if ChatGPT can differentiate the quality difference between different system outputs for the same source document.

In fact, we find that Pearson's $r$ calculated by Method 1 and Method 2 are highly correlated. In Table 4, we show the result of Topical-Chat while we use Method 2 to calculate Pearson's $r$; Kendall's $\tau$ is still calculated by Method 2. Comparing the results of Pearson's $r$ in Table 2 and Table 4, one can easily see that when a method have significantly higher Pearson's $r$ in Table 2, it will also have significantly higher Pearson's $r$. We present the $r$ calculated by Method 1 because it makes more sense when calculating statistical significance when the correlation coefficient is calculated at the dataset-level (Graham et al., 2015).

## D Results of Changing the Temperature and Prompts

We show the results of varying the temperature used to sample the ChatGPT output in Table 5. In the experiments in this section, we only sample $N = 5$ samples from the ChatGPT since we find that G-eval and our proposed guidelines are quite robust to the number of samples when $N \geq 5$.

## E Datasets

### E.1 SummEval

SummEval (Fabbri et al., 2021) is a dataset for the meta-evaluation of summarization. It contains 100 source documents, each with 16 summaries obtained from different summarization models. Each of the 1600 summaries is rated by three workers recruited on Amazon Mturk and two experts in summarization. Each summary in SummEval is rated by humans based on the *coherence*, *consistency*, *fluency* of the summary, and *relevance* between the summary and the source document. Each attribute is rated based on a 5-point Likert scale.

We download the source documents, summaries, and human ratings from the GitHub repository of G-Eval (`https://github.com/nlpyang/geval/tree/8f54105/data`). SummEval was released under MIT License, and our usage for research does not violate the dataset's initial intention.

### E.2 Topical-Chat

Topical-Chat (Gopalakrishnan et al., 2019) is a knowledge-grounded open-domain dialogue dataset. The dataset consists of a dialogue context (history), an interesting fact related to the topic of the conversation, and a response. Mehri and Eskenazi (2020) releases high-quality human annotations on the quality of responses. They construct the dataset as follows: they first sample 60 dialogues context from Topical-Chat, and for each dialogue context and corresponding fun fact, they use a transformer model to generate four responses using four decoding methods. Each dialogue content has two additional responses: the human response and the ground truth response. Thus, there are a total of 360 dialogue-response pairs. Those pairs are evaluated based on six attributes, and we follow Zhong et al. (2022) and Liu et al. (2023) to only use four attributes: *naturalness*, *coherence*, *engagingness*, and *groundedness* (whether the response is grounded on the provided knowledge). We obtain the human ratings of Topical-Chat from the Github repository of UniEval (Zhong et al., 2022): `https://github.com/maszhongming/UniEval/blob/main/reproduce/data/dialogue/topical_chat.json`.

## F Prompts

We list the prompts we use in this section. In the main content of the paper and in the following parts, we use different highlight colors to represent different parts of the prompt. A prompt is composed

| Sec. | Ablations | | Naturalness | | Coherence | | Engagingness | | Groundedness | |
|---|---|---|---|---|---|---|---|---|---|---|
| | CoT | Output | $r$ | $\tau$ | $r$ | $\tau$ | $r$ | $\tau$ | $r$ | $\tau$ |
| GPT-4[†] | ✓ | *Score only* | 0.549 | - | 0.594 | - | 0.627 | - | 0.531 | - |
| 3.1 | ✓ | *Score only* | 0.445 | 0.358 | 0.498 | 0.391 | 0.579 | 0.513 | 0.685 | 0.566 |
| | ✗ | | 0.431 | 0.331 | 0.507 | 0.404 | 0.631 | 0.535 | 0.666 | 0.582 |
| 3.2 | ✗ | *Score only* | 0.431 | 0.331 | 0.507 | 0.404 | 0.631 | 0.535 | 0.666 | 0.582 |
| | ✗ | *Free Text* | 0.572 | 0.476 | 0.523 | 0.426 | 0.676 | 0.557 | 0.747 | 0.666 |
| | ✗ | *Rate-explain* | 0.621 | 0.512 | 0.472 | 0.425 | 0.61 | 0.509 | 0.771 | 0.663 |
| | ✗ | *Analyze-rate* | 0.573 | 0.47 | 0.486 | 0.416 | 0.628 | 0.524 | 0.725 | 0.693 |

Table 4: The Pearson's $r$ and Kendall's $\tau$ correlation coefficient between LLMs' ratings and human ratings for Topical-Chat. Note that in this table, both Pearson's $r$ and Kendall's $\tau$ are calculated by Method 2 in Appendix C.0.2. All the results in this table, except the first row, are from ChatGPT. The results of GPT-4 are from Liu et al. (2023) but should not be compared with our results since the prompts they use may be different from the prompt we use. Still, we can see that for *naturalness*, *engagingness*, and *grounedness*, the results of *rate-explain* and *analyze-rate* is better or comparable to GPT-4.

of four parts: (1) the **descriptions of the rating task**, (2) the **definition and rating criteria** of the attribute to be rated, (3) the **sample to be rated**, and (4) **a sentence used to prompt the LLM to give the rating**.

The prompts for different attributes of the same dataset share the same descriptions of the rating task. Different attributes use different definition and rating criteria. In G-Eval, the prompts also compose of the evaluation steps generated by auto CoT.

### F.1 Prompts for SummEval

The **descriptions of the rating task**, the **definition and rating criteria**, the evaluation steps for *coherence*, *consistency*, and *relevance* in SummEval is from the prompts released by G-Eval in their GitHub repository (https://github.com/nlpyang/geval/tree/8f54105/prompts/summeval). While G-Eval also releases the prompt they use for *fluency*, we find something **highly problematic** in the prompt they use. The prompt for fluency asks the LLM to rate fluency **on a scale of 1 to 3** (https://github.com/nlpyang/geval/blob/8f54105061e00377fbb909153892d5bfb5b3623a/prompts/summeval/flu_detailed.txt), while the original rating scale in SummEval is **1 to 5**. We also find that the original rating criteria used in G-Eval for fluency differ largely from the rating criteria of fluency used for human evaluation in SummEval. Through our experiment, we find that the misalignment of evaluation criteria and evaluation scale significantly decreases Pearson's $r$ with human ratings when using *analyze-rate* to

prompt ChatGPT to output. This is likely because ChatGPT tends to stick to the rating criteria when prompted with *analyze-rate*, and when using the rating criteria different from the criteria that are used to instruct the human raters, the scores generated by ChatGPT deviates more from the human ratings. This highlights the importance of using the same instructions to the LLM as those instructions used in the human evaluation, as emphasized in Chiang and Lee (2023).

First, we show an example prompt for *coherence*. This prompt corresponds to the *score only + auto CoT* in Table 1.

**Coherence**

You will be given one summary written for a news article.
Your task is to rate the summary on one metric.
Please make sure you read and understand these instructions carefully. Please keep this document open while reviewing, and refer to it as needed.
Evaluation Criteria:
Coherence (1-5) – the collective quality of all sentences. We align this dimension with the DUC quality question of structure and coherence whereby "the summary should be well-structured and well-organized. The summary should not just be a heap of related information, but should build from sentence to a coherent body of information about a topic."
Evaluation Steps:
1. Read the news article carefully and

| Auto-CoT | Output | Coherence | Consistency | Fluency | Relevance |
|:---:|:---:|:---:|:---:|:---:|:---:|
| ✓ | *Score only* | 0.356 | 0.290 | 0.261 | 0.263 |
| ✗ | *Rate-explain* | **0.548** | **0.482** | **0.423** | **0.487** |
| ✗ | *Analyze-rate* | **0.589** | **0.439** | **0.438** | **0.319** |

(a) Temperature $T = 0.3$

| Auto-CoT | Output | Coherence | Consistency | Fluency | Relevance |
|:---:|:---:|:---:|:---:|:---:|:---:|
| ✓ | *Score only* | 0.394 | 0.256 | 0.288 | 0.334 |
| ✗ | *Rate-explain* | **0.526** | **0.468** | **0.414** | **0.485** |
| ✗ | *Analyze-rate* | **0.605** | **0.448** | **0.441** | **0.392** |

(b) Temperature $T = 0.7$

| Auto-CoT | Output | Coherence | Consistency | Fluency | Relevance |
|:---:|:---:|:---:|:---:|:---:|:---:|
| ✓ | *Score only* | 0.450 | 0.370 | 0.319 | 0.403 |
| ✗ | *Rate-explain* | **0.557** | **0.473** | **0.452** | **0.509** |
| ✗ | *Analyze-rate* | **0.635** | **0.534** | **0.479** | **0.444** |

(c) Temperature $T = 1.0$ (The result in Table 1)

Table 5: Comparing G-Eval (Auto-CoT + score only) with *rate-explain* and *analyze-rate* at different temperatures. We boldface Pearson's r statistically significantly higher than the baseline (the first row in each subtable).

| Auto-CoT | Output | Coherence | Consistency | Fluency | Relevance |
|:---:|:---:|:---:|:---:|:---:|:---:|
| ✓ | *Score only* | 0.308 | 0.248 | 0.265 | 0.345 |
| ✗ | *Rate-explain* | textbf0.526 | **0.468** | **0.414** | **0.485** |
| ✗ | *Analyze-rate* | **0.589** | **0.524** | **0.459** | 0.416 |

(a) Results when prompted with the human evaluator prompts.

| Auto-CoT | Output | Coherence | Consistency | Fluency | Relevance |
|:---:|:---:|:---:|:---:|:---:|:---:|
| ✓ | *Score only* | 0.325 | 0.206 | 0.281 | 0.301 |
| ✗ | *Rate-explain* | **0.596** | **0.465** | 0.403 | **0.478** |
| ✗ | *Analyze-rate* | **0.596** | **0.493** | **0.475** | 0.406 |

(b) Results when prompted with the HHH prompts.

Table 6: Comparing G-Eval (Auto-CoT + score only) with *rate-explain* and *analyze-rate* when using different prompts. We boldface Pearson's r statistically significantly higher than the baseline (the first row in each subtable).

```
identify the main topic and key points.
2. Read the summary and compare it to the
news article. Check if the summary cov
ers the main topic and key points of the
news article, and if it presents them in
a clear and logical order.
3. Assign a score for coherence on a scale
of 1 to 5, where 1 is the lowest and 5
is the highest based on the Evaluation
Criteria.
Example:
Source Text: {{Document}}
Summary: {{Summary}}
Evaluation Form (scores ONLY):
- Coherence:
```

### F.1.1 Different Output Prompts

For different output prompts, which is the ablation in Section 3.2 and the last block in Table 1 and 2, we only change the yellow parts (the last part) in the example prompt above. There are four output prompts used in Section 3.2: *score only*, *free text*, *rate-explain*, and *analyze-rate*. The prompts for *free text* is attribute-dependent, and we list them in the Their corresponding output prompts are listed as follows:

**Score only**
```
 Evaluation Form (scores ONLY):
- {Attribute}:
```

**Rate-explain**
```
 Evaluation Form (Answer by starting with
"Rating:" and then give the explanation
of the rating on the next line by "Ratio
nale:"):
- {Attribute}:
```

**Analyze-rate**
```
 Evaluation Form (Answer by starting with
"Analysis:" to analyze the given example
regarding the evaluation criteria as con
cise as possible, and then give the nu
meric rating on the next line by "Rat
ing:):
- {Attribute}:
```

### F.1.2 Attribute-Dependent Prompts

The definition and rating criteria of the attribute to be rated, the evaluation steps generated by auto CoT, and output prompt for *text-free* are attribute-dependent, and we list them as follows. We use different colors to denote different parts in the prompt.

Note that the following prompts are not the complete prompts used as the model input; they need to be used with the descriptions of the rating task and the sample to be rated.

**Coherence**
```
Evaluation Criteria:
Coherence (1-5) - the collective quality
of all sentences. We align this dimen
sion with the DUC quality question of
structure and coherence whereby "the
summary should be well-structured and
well-organized. The summary should not
just be a heap of related information,
but should build from sentence to a
coherent body of information about a
topic."

Evaluation Steps:
1. Read the news article carefully and
identify the main topic and key points.
2. Read the summary and compare it to
the news article. Check if the summary
covers the main topic and key points of
the news article, and if it presents them
in a clear and logical order.
3. Assign a score for coherence on a
scale of 1 to 5, where 1 is the lowest and
5 is the highest based on the Evaluation
Criteria.

Question:
How coherent is the summary? That is,
how well do the sentences in the summary
fit together? (On a scale of 1-5, with 1
being the lowest)
```

**Consistency**
```
Evaluation Criteria:
Consistency (1-5) - the factual alignment
between the summary and the summarized
source. A factually consistent summary
contains only statements that are en
tailed by the source document. Annotators
were also asked to penalize summaries
that contained hallucinated facts.

Evaluation Steps:
1. Read the news article carefully and
identify the main facts and details it
presents.
2. Read the summary and compare it to the
```

article. Check if the summary contains any factual errors that are not supported by the article.
3. Assign a score for consistency based on the Evaluation Criteria.

Question:
How consistent is the summary with the source document in terms of the factual alignment? (On a scale of 1-5, with 1 being the lowest)

**Fluency**
Evaluation Criteria:
Fluency (1-5): This rating measures the quality of individual sentences, are they well-written and grammatically correct. Consider the quality of individual sentences.

Evaluation steps:
1. Read the given summary.
2. Evaluate the fluency of the summary on a scale of 1-5 based on the criteria provided.
3. Provide the rating.

Question:
Based on the evaluation criteria, how fluent is the summary? (On a scale of 1-5, with 1 being the lowest)

**Relevance**
Evaluation Criteria:
Relevance (1-5) - selection of important content from the source. The summary should include only important information from the source document. Annotators were instructed to penalize summaries which contained redundancies and excess information.

Evaluation Steps:
1. Read the summary and the source document carefully.
2. Compare the summary to the source document and identify the main points of the article.
3. Assess how well the summary covers the main points of the article, and how much irrelevant or redundant information it contains.

4. Assign a relevance score from 1 to 5.

Question:
On a scale of 1-5, with 1 being the lowest, is the summary relevant to the source document and does the summary only contain the important information of the source document?

## F.2 Prompts for Topical-Chat

First, we show an example prompt for *naturalness*. This prompt corresponds to the *score only + auto CoT* in Table 2.

**Naturalness**
You will be given a conversation between two individuals. You will then be given one potential response for the next turn in the conversation. The response concerns an interesting fact, which will be provided as well.
Your task is to rate the responses on one metric.
Please make sure you read and understand these instructions carefully. Please keep this document open while reviewing, and refer to it as needed.

Evaluation Crieteria:
Naturalness (1-3) Is the response naturally written??
- A score of 1 (bad) means that the response is unnatural.
- A score of 2 (ok) means the response is strange, but not entirely unnatural.
- A score of 3 (good) means that the response is natural.

Evaluation Steps:
1. Read the conversation between the two individuals.
2. Read the potential response for the next turn in the conversation.
3. Evaluate the response based on its naturalness, using the provided criteria.
4. Assign a rating score of 1, 2, or 3 based on the evaluation.

Example:
Conversation History:
{{Document}}
Corresponding Fact:

```
{{Fact}}
Response:
{{Response}}

Evaluation Form (scores ONLY):
- Naturalness:
```

### F.2.1 Different Output Prompts

For Topical-Chat, we also conduct ablations on different output prompts. Those different output prompts for *score only*, *rate-explain*, *analyze-rate* are the same as those listed in Section F.1.1. We do not list them here to save some space. The exact prompts we use can be found in the supplementary data of this paper.

### F.2.2 Attribute-Dependent Prompts

The definition and rating criteria of the attribute to be rated, the evaluation steps generated by auto CoT, and output prompt for *text-free* are attribute-dependent, and we list them as follows. Again, the following prompts are not the complete prompts used as the model input; they need to be used with the descriptions of the rating task and the sample to be rated.

**Naturalness**
```
Evaluation Crieteria:
Naturalness (1-3) Is the response natu
rally written??
- A score of 1 (bad) means that the
response is unnatural.
- A score of 2 (ok) means the response
is strange, but not entirely unnatural.
- A score of 3 (good) means that the
response is natural.

Evaluation Steps:
1. Read the conversation between the two
individuals.
2. Read the potential response for the
next turn in the conversation.
3. Evaluate the response based on its
naturalness, using the provided criteria.
4. Assign a rating score of 1, 2, or 3
based on the evaluation.

Question:
How natural is the reponse? (On a scale
of 1-3, with 1 being the lowest)
```

**Coherence**
```
Evaluation Crieteria:
```

```
Coherence (1-3) Does the response serve
as a valid continuation of the conversa
tion history?
- A score of 1 (no) means that the
response drastically changes topic or
ignores the conversation history.
- A score of 2 (somewhat) means the
response refers to the conversation
history in a limited capacity (e.g., in a
generic way) and shifts the conversation
topic.
- A score of 3 (yes) means the response
is on topic and strongly acknowledges
the conversation history.

Evaluation Steps:
1. Read the conversation history.
2. Read the potential response.
3. Evaluate the coherence of the response
based on the conversation history.
4. Assign a score of 1, 2, or 3 for
coherence.

Question:
Does the response serve as a valid con
tinuation of the conversation history?
(On a scale of 1-3, with 1 meaning the
response is invalid and 3 meaning the
response is coherent)
```

**Engagingness**
```
Evaluation Crieteria:
Engagingness (1-3) Is the response dul
l/interesting?
- A score of 1 (dull) means that the
response is generic and dull.
- A score of 2 (somewhat interesting)
means the response is somewhat inter
esting and could engage you in the
conversation (e.g., an opinion, thought)
- A score of 3 (interesting) means the
response is very interesting or presents
an interesting fact

Evaluation Steps:
1. Read the conversation, the correspond
ing fact and the response carefully.
2. Rate the response on a scale of
1-3 for engagingness, according to the
criteria above.

Question:
```

Is the response interesting and engaging? (On a scale of 1-3, with 1 meaning dull and 3 meaning interesting)

**Groundedness**

Evaluation Crieteria:
Groundedness (0-1) given the fact that this response is conditioned on, deter mine whether this response uses that fact.
- A score of 0 (no) means the response does not mention or refer to the fact at all
- A score of 1 (yes) means the response uses the fact well

Evaluation Steps:
1. Read the conversation between the two individuals.
2. Identify the fact that is provided for the potential response.
3. Read the potential response.
4. Determine if the potential response uses or mentions the fact.
5. Assign a score of 0 or 1 for grounded ness based on whether the response uses the fact.

Question:
Given the fact that this response is conditioned on, does the response use the fact? (On a scale of 0-1, with 0 meaning no and 1 meaning yes)

### F.3 Prompts for Section 3.4.2

**HHH prompts**     You are an AI assistant. The AI tries to be helpful, polite, honest, sophisticated, emotionally aware, and humble-but-knowledgeable. The assistant is happy to help with almost anything, and will do its best to understand exactly what is needed.

**Human annotator prompts**     Assume that you are a professional and careful human evaluator. You are recruited and paid to conduct the following task. You need to strictly follow the task instruction and ensure that you are doing the job with high-quality.