# OpenReview forum: "A Closer Look into Using Large Language Models for Automatic Evaluation"
_EMNLP/2023/Conference — EMNLP 2023 Findings_

### Official Review · Reviewer_f4MA · 2023-08-04

**Soundness:** 3

**Excitement:**

4: Strong: This paper deepens the understanding of some phenomenon or lowers the barriers to an existing research direction.

**Paper Topic And Main Contributions:**

The paper investigates whether large language models (LLMs) could be used to evaluate the quality of a text. Previous studies have demonstrated that LLMs could be prompted to generate ratings across different aspects of text quality. In this work, the authors perform a deeper analysis of the existing work and provide recommendations. For example, they observe that making LLMs to output only a single numeric rating is suboptimal. The authors also propose additional prompting strategies inspired by chain-of-thought prompting, which further improves the results in terms of the alignment of ratings generated by LLMs to human judgement.

**Questions For The Authors:**

Given a text, did you generate the ratings multiple times? When performing an experiment, multiple annotators are asked to give their ratings, which might also differ as the ratings might be subjective. I was just wondering if it is the same for LLMs? Maybe they generate slightly different ratings each time.

Did you also consider open-source models such as LLaMa?

**Reasons To Accept:**

Explores an interesting problem of whether LLMs could be utilized to rate text quality. Rating text quality is an expensive process and involves employing human annotators. If LLMs are able to mimic human evaluators, it would be a much cheaper option.

Provides interesting recommendations regarding the usage of LLMs for text rating.

Proposes additional prompt strategies to improve the performance.

Experiments performed on real-world datasets.

**Reasons To Reject:**

As has been pointed out in the limitation section, then content of the paper is too big to fit in a short paper. One has to often refer to the appendix while reading it. I believe that the main paper should self-contained, which is not in this case.



**Reproducibility:**

4: Could mostly reproduce the results, but there may be some variation because of sample variance or minor variations in their interpretation of the protocol or method.

**Reviewer Confidence:**

4: Quite sure. I tried to check the important points carefully. It's unlikely, though conceivable, that I missed something that should affect my ratings.

---

> ### Author Rebuttal · Authors · 2023-08-28
>
> Thank you for your positive and thoughtful reviews. We are grateful that the reviewer considers our paper interesting and understands the significance of our work. In the rebuttal to reviewer Efov, we conduct further analysis to support our guidelines better. This definitely makes our paper stronger. We hope the reviewer can help us publish our paper in EMNLP 2023.  We respond to your review as follows.
>
>
> # Response to Reasons to Reject
>
>
> > As has been pointed out in the limitation section, then content of the paper is too big to fit in a short paper. One has to often refer to the appendix while reading it. I believe that the main paper should self-contained, which is not in this case.
>
>
> - We will modify our paper using the additional page in the camera-ready version to address the issue. We will move Table 2 (results for Topical-Chat) to the main content, and we will explain the meaning of the colours in Section 2.1. We will also illustrate some of the full prompts (currently located in Appendix E) to the main content for better understanding. We are willing to make further improvements based on the reviewer's feedback!
>
>
> # Reponse to Questions
> > Given a text, did you generate the ratings multiple times? When performing an experiment, multiple annotators are asked to give their ratings, which might also differ as the ratings might be subjective. I was just wondering if it is the same for LLMs? Maybe they generate slightly different ratings each time.
>
>
> - In our experiment, we follow Liu et al. (2023) to sample 20 responses from ChatGPT. We left this detail in Line 566 in the Appendix. Still, we find that using as few as five sampled responses is enough to achieve high correlations with human ratings. We will add the above discussion in Section 2.3.
>
>
>
>
> > Did you also consider open-source models such as LLaMa?
>
>
> - We did not use open-source models since, at the time of writing the paper, we found that the best instruction-following model on Open LLM Leaderboard (Falcon-40b Instruct, which is better than LLaMA) cannot follow the instructions to perform evaluation very well, so we omit open-source LLMs in our study. The above discussions are included in Section 2.3, and we will also add this to the limitation part of our paper. (Note that LLaMA 2 was unavailable when we submitted this paper to EMNLP 2023.)

---

### Official Review · Reviewer_Efov · 2023-08-05

**Soundness:** 3

**Excitement:**

3: Ambivalent: It has merits (e.g., it reports state-of-the-art results, the idea is nice), but there are key weaknesses (e.g., it describes incremental work), and it can significantly benefit from another round of revision. However, I won't object to accepting it if my co-reviewers champion it.

**Paper Topic And Main Contributions:**

The paper compares two recently introduced LLM-based evaluation systems: G-eval and LLM evaluation. It's contributions are as follows:
1. It is not optimal to always make the LLMs output a single numeric number, but asking the LLMs to reason it's rating improves the correlation with human evaluaion
2. Auto-Chain of Thought reasoning does not increase the correlation between automatic and human evaluation,
3. Using ChatGPT as the LLM and by making LLMs rationalize its rating, the paper achieves higher correlation with human metrics on two datasets

**Questions For The Authors:**

- Could "LLM evaluation" be called something else? It leads to unnecessary confusion in parts of the text


**Reasons To Accept:**

1. The paper studies the differences between two popular LLM-based auto evaluation metrics and systematically studies questions such as  "should auto-COT be always used?"
2. The study recommends practices that can be potentially useful for automatic evaluation
3. The paper is well presented (except for a few text highlights) and is easy to follow

**Reasons To Reject:**

1. While this work is relevant, it has been only limited to ChatGPT, making the generalization to other LLMs questionable
2. Considering the performance of LLMs are sensitive to prompts, type of LLM model, temperature parameter etc., more empirical evidence is needed to concretely support the guidelines

**Reproducibility:**

5: Could easily reproduce the results.

**Reviewer Confidence:**

5: Positive that my evaluation is correct. I read the paper very carefully and I am very familiar with related work.

**Typos Grammar Style And Presentation Improvements:**

- Why are the lines 83 to 102 (and some other parts) are coloured? Could you remove the highlighting of words in the text?

---

> ### Author Rebuttal · Authors · 2023-08-28
>
> Thank the reviewer for the detailed reviews and the concerns you raised. Addressing your concerns surely makes our paper much better and more sound. We respond to your review as follows.
>
>
> # Summary of Responses (TL;DR)
>
> 1. We conduct further experiments on two datasets by changing the temperature and prompts, and we find that our main guideline (*asking the LLM to explain/analyze outperforms the method proposed in G-Eval (using score only + auto-CoT)*) always holds. We also find that the correlations obtained by asking the LLM to explain/analyze are much more stable when varying the sampling temperatures and prompts (compared with G-Eval). Thus, **our guideline is concretely supported by all the experiments.**
> 2. We explain why we only use ChatGPT. We argue that our results are useful since we achieve a much better performance using a highly accessible model (ChatGPT). The NLP community can benefit from our results.
>
> We believe the above responses and additional experiments largely address the reviewer's concerns. We hope the reviewer can kindly adjust the ratings to help us publish this paper.
>
> # Response to Reasons to Reject
>
> > While this work is relevant, it has been only limited to ChatGPT, making the generalization to other LLMs questionable
>
> - We are grateful that the reviewer considers our work relevant, and we think that only using ChatGPT is not a significant weakness of our paper for two reasons. (1) The goal of the paper is to understand previous evaluations using LLMs, and the two prior works both use ChatGPT. We thus limit our study to ChatGPT. (2) Our goal is to use LLMs to conduct **better** evaluations. Our results show that using ChatGPT, when asking it to explain its rating, can achieve much better alignment with humans and sometimes achieve SoTA alignment with human ratings. This result is significant and valuable as it improves the current state of automatic evaluation. We believe such a result is attractive and usable to the NLP community.
> - We only use ChatGPT because this is the only model that is affordable and open enough for most people to use, compared with PaLM (not publicly availble), Claude (only available in USA and UK), or GPT-4 (do not provide API access at the time of June 2023). We do not use other open-source models since we find them unable to follow the task instructions very well. We explain the choice of LLM in Section 2.3.
>
> > Considering the performance of LLMs are sensitive to prompts, type of LLM model, temperature parameter etc., more empirical evidence is needed to concretely support the guidelines
>
> Thank the reviewer for this great question! The main guideline we propose is "asking the LLM to explain/analyze outperforms the method proposed in G-Eval (using score only + auto-CoT)". In the following supplementary experiments, we conduct further experiments on SummEval and Topical Chat to verify the robustness of the guideline. **The conclusion from Topical Chat is identical to the conclusion obtained from SummEval, and we are only showing the results of SummEval here for simplicity.**
>
> ### **Takeaway**
> The takeaway from the following experiments is that asking the LLM to explain/analyze still outperforms G-Eval when varying the temperature used in sampling and changing the prompts to ChatGPT. We also find that the performance of G-Eval is very sensitive to the sampling temperature, while the correlations obtained by explain/analyze fluctuate less when varying the temperature. This shows that asking ChatGPT to explain/analyze when rating is more robust than requiring the LLM to output only a number. We will add the following experiments in our appendix and shortly discuss this in the main content in our revision.
>
>
> ### **Ablation 1: Different temperature $T$ during generation**
> We check if our guideline holds if we change the temperature during generation. We compare the Pearson's *r* when using the method proposed in G-Eval (Auto-CoT + score only) with *Rate-explain* and *Analyze-rate* under different temperatures used when generating the output from ChatGPT. We follow Chiang and Lee (2023) and use two temperatures: 0.7 (shown in Rebuttal Table 1) and 0.3 (shown in Rebuttal Table 2). We also recap the results in our paper in Rebuttal Table 3, where the temperature is set to 1.0. The ChatGPT we use during rebuttal is `gpt3.5-turbo-0613`, which is the same model we used in the experiments in our paper.
>
> First, when fixing the sampling temperature, we can see that *Rate-explain* and *Analyze-rate* always achieve a higher correlation compared with G-Eval (the first row in Rebuttal Table 1, 2, and 3). This supports our guideline that *"asking the LLM to explain/analyze outperforms the method proposed in G-Eval"*.
> Next, we observe that the correlation of G-Eval when $T=0.3$ (the first row in Rebuttal Table 2) is much lower than the correlation of $T=1.0$ (the first row in Rebuttal Table 3). This shows that G-Eval is not robust to sampling temperature. However, by comparing Rebuttal Table 1, 2, and 3, we find that the correlations obtained by *Rate-explain* and *Analyze-rate* do not significantly change for different sampling temperatures for almost all cases. **This shows that *Rate-explain* and *Analyze-rate* is more robust than G-Eval with respect to the sampling temperature.**
>
>
> **Rebuttal Table 1. Comparing G-Eval (Auto-CoT + score only) with Rate-explain and Analyze-rate when the temperature of generation is set to 0.7. We boldface the Pearson’s r statistically significantly higher than the baseline (the first row).**
>
> |Auto-CoT | Output | Temperature |Coherence| Consistency| Fluency | Relevance |
> |:-:|:-:|:-:|:-:|:-:|:-:|:-:|
> |v | Score only | 0.7 | 0.394 | 0.256| 0.288 | 0.334 |
> |x |Rate-explain | 0.7 | **0.526** | **0.468**| **0.414** | **0.485** |
> |x |Analyze-rate | 0.7 | **0.605** | **0.448**| **0.441** | **0.392** |
>
>
> **Rebuttal Table 2. Comparing G-Eval (Auto-CoT + score only) with Rate-explain and Analyze-rate when the temperature of generation is set to 0.3. We boldface the Pearson’s r statistically significantly higher than the baseline (the first row).**
>
> |Auto-CoT | Output | Temperature |Coherence| Consistency| Fluency | Relevance|
> |:-:|:-:|:-:|:-:|:-:|:-:|:-:|
> |v | Score only | 0.3 | 0.356 | 0.290 | 0.261 | 0.263 |
> |x |Rate-explain | 0.3 | **0.548** | **0.482**| **0.423** | **0.487** |
> |x |Analyze-rate | 0.3 | **0.589** | **0.439**| **0.438** | **0.319** |
>
>
> **Rebuttal Table 3. Comparing G-Eval (Auto-CoT + score only) with Rate-explain and Analyze-rate when the temperature of generation is set to 1.0 (The result in our paper). We boldface the Pearson’s r statistically significantly higher than the baseline (the first row).**
>
> |Auto-CoT | Output | Temperature |Coherence| Consistency| Fluency | Relevance|
> |:-:|:-:|:-:|:-:|:-:|:-:|:-:|
> |v | Score only | 1.0 | 0.450 | 0.370 | 0.319 | 0.403 |
> |x |Rate-explain | 1. 0|  **0.557** | **0.473** | **0.452** | **0.509** |
> |x |Analyze-rate | 1.0 | **0.635** | **0.534** | **0.479** | **0.444** |
>
> ---
>
> ### **Ablation 2: Different prompts to ChatGPT during generation**
> We check if our guideline holds if we change the prompt given to ChatGPT. In this experiment, we changed the prompts to ChatGPT by appending some instructions before the descriptions of the rating task.
> We tried with two prompts: (1) the **HHH prompts**, as detailed below
> ```
> You are an AI assistant.
> The AI tries to be helpful, polite, honest, sophisticated, emotionally aware, and humble-but-knowledgeable.
> The assistant is happy to help with almost anything, and will do its best to understand exactly what is needed.
> ```
> and (2) **human annotator prompts**, as shown below
> ```
> Assume that you are a professional and careful human evaluator.
> You are recruited and paid to conduct the following task.
> You need to strictly follow the task instruction and ensure that you are doing the job with high-quality.
> ```
> The HHH prompt is designed by Bai et al. (2022) to align the output of LLMs to be more harmless, honest, and helpful. The human annotator prompt is inspired by Chiang and Lee (2022),  who use a similar prompt to make the LLM behave as a human annotator.
> These two prompts will be inserted in front of the prompt we originally used in our paper. (Precisely, we insert either the HHH prompt or the human-evaluator prompt before Line 734 on page 9.) We use these two prompts to inject persona into the LLM. This is inspired by Zeng et al. (2022), which shows that the output of GPT3 can be different when prompted with different persona.
> We use temperature $T=1.0$ in this expeirment. The results are shown in the Rebuttal Table 4 and 5.
>
> We find that ***Rate-explain*** **and** ***Analyze-rate*** **consistently outperform the G-eval (auto-CoT + score only)** when using the human annotator prompts and the HHH prompts. **The results indicates that our guidelines is robust toward different prompts**. By comparing the results in Rebuttal Table 4 and 5 with the results in Table 1 in the paper, we also find that the correlations of G-Eval (the second row in Table 1) significantly drop when adding the human-annotator prompts or the HHH prompts. On the other hand, the correlation for *Rate-explain* and *Analyze-rate* do not show to such significance decreases when adding the human-annotator prompt and the HHH prompt. **This shows that asking the LLM to explain is more robust to the variation of the prompts.**
>
> **Rebuttal Table 4. Comparing G-Eval (Auto-CoT + score only) with Rate-explain and Analyze-rate when using the human annotator prompt. We boldface the Pearson’s r statistically significantly higher than the baseline (the first row).**
>
> |Auto-CoT | Output | Additional prompt |Coherence| Consistency| Fluency | Relevance|
> |:-:|:-:|:-:|:-:|:-:|:-:|:-:|
> |v | Score only | human annotator prompt | 0.308 | 0.248 | 0.265 | 0.345 |
> |x |Rate-explain | human annotator prompt| **0.526** | **0.468**| **0.414** | **0.485** |
> |x |Analyze-rate | human annotator prompt | **0.589** | **0.524** | **0.459** | **0.416** |
>
>
> **Rebuttal Table 5. Comparing G-Eval (Auto-CoT + score only) with Rate-explain and Analyze-rate when using the HHH prompt. We boldface the Pearson’s r statistically significantly higher than the baseline (the first row).**
>
> |Auto-CoT | Output | Additional prompt |Coherence| Consistency| Fluency | Relevance|
> |:-:|:-:|:-:|:-:|:-:|:-:|:-:|
> |v | Score only | HHH prompt | 0.325 | 0.206 | 0.281 | 0.301|
> |x |Rate-explain |HHH prompt| **0.596** | **0.465** | **0.403** | **0.478** |
> |x |Analyze-rate | HHH prompt | **0.596** | **0.493** | **0.475** | **0.406** |
>
> # Response to Questions
>
>
> > Could "LLM evaluation" be called something else? It leads to unnecessary confusion in parts of the text
>
> “LLM evaluation” is the name proposed by Chiang and Lee (2023). We will change it into "*LLM-Eval*" in our revision for clearance.
>
> # Responses to Presentation Improvements
>
>
> > Why are the lines 83 to 102 (and some other parts) are coloured? Could you remove the highlighting of words in the text?
>
> In the main content of the paper, we use different highlight colors to represent different parts of the prompt. We use (1) red for the task description, (2) green for the definition and rating criteria, (3) yellow for the sentence used to prompt the LLM to give the rating, and (4) blue for the evaluation steps generated by Auto-CoT. The coloring is supposed to make the readers easier to distinguish different parts in the prompts. We will explicitly explain this in the main content of the paper in the camera-ready version. However, if the reviewer thinks the coloring is unnecessary and confusing, we can remove those colorings.
>
>
> References
> ===
> Bai, Yuntao, et al. "Training a helpful and harmless assistant with reinforcement learning from human feedback." arXiv preprint arXiv:2204.05862 (2022).
>
> Zeng, Andy, et al. "Socratic models: Composing zero-shot multimodal reasoning with language." arXiv preprint arXiv:2204.00598 (2022).

---

### Official Review · Reviewer_fsws · 2023-08-07

**Soundness:** 3

**Excitement:**

3: Ambivalent: It has merits (e.g., it reports state-of-the-art results, the idea is nice), but there are key weaknesses (e.g., it describes incremental work), and it can significantly benefit from another round of revision. However, I won't object to accepting it if my co-reviewers champion it.

**Paper Topic And Main Contributions:**

This paper deals with the task of exploring text quality using LLMs. Authors use LLM Evaluation and G-eval frameworks and compare this with human ratings. There are some important findings, like e.g. the Auto Chain-of-Thought of Geval does not always seem to correlate best with human ratings and also that reducing outputs to a numeric value seems to not be ideal since LLMs can explain their ratings and that adds to explainabilty.

**Questions For The Authors:**

A) I would expand the limitations section on potential misuses (perhaps in conjuction with the ethical analysis)
B) How robust you think the results are given the analysis? What are your strongest conclusions?

**Reasons To Accept:**

1) Clear research questions
2) Clear experimental procedure which is important in the LLM era.

**Reasons To Reject:**

1) Evaluation of CoT seems not robust, given the part release of prompts. It's mentioned in the limitations, but I am not sure how useful are the results in the end.
2) Limited evaluation of results/tasks and details on how to reproduce them. To be fair, this is a general remark for all the prompting papers.
3) More targeted conclusions are missing, e.g. recommendations for further prompting.

**Reproducibility:**

4: Could mostly reproduce the results, but there may be some variation because of sample variance or minor variations in their interpretation of the protocol or method.

**Reviewer Confidence:**

3: Pretty sure, but there's a chance I missed something. Although I have a good feel for this area in general, I did not carefully check the paper's details, e.g., the math, experimental design, or novelty.

---

> ### Author Rebuttal · Authors · 2023-08-28
>
> Thank you for the insightful reviews and for considering our research question and experiment to be clear. Addressing your concerns helps us improve our paper. We respond to your review as follows.
>
> Response to Reasons to Reject
> ===
> > Evaluation of CoT seems not robust, given **the part release of prompts**. It's mentioned in the limitations, but I am not sure **how useful are the results in the end**.
>
> - **The part release of prompts**: We release all the prompts we used in Appendix E, instead of releasing partial prompts.
> - **Evaluation of CoT seems not robust**: In our experiments, we sample $N=20$ CoT reasoning paths for each rating attribute and for each sample from ChatGPT and average the ratings. We find that varying $N$ (the number of sampled CoT paths) from $N=5$ to $N=20$ merely changes the conclusion in the paper. In the response to reviewer Efov, we also show the results when we vary the temperature for sampling and prompts when asking ChatGPT to rate the samples. We find that, again, the results merely vary: asking the ChatGPT to explain/analyze achieves the best correlation with human ratings. **This shows that CoT (*Rate-explain* and *Analyze-rate*) are quite robust, so our results are very useful.** Please find the details on the experiments in the rebuttal to reviewer Efov.
>
> > Limited evaluation of results/tasks and details on how to reproduce them. To be fair, this is a general remark for all the prompting papers.
>
> - **Limited details on how to reproduce the results**: All the prompts we use are listed in Appendix E. We also specify the models we use in Section 2.3, and all the details about the datasets, including how to download them, are described in Appendix D. One can easily reproduce our experiments based on the information in our paper. We will release all the source codes and prompts we use on GitHub upon acceptance.
> - **Limited results/tasks**: We conduct our experiments on two types of NLP tasks (summarization and dialogue generation), following G-Eval (Liu et al. 2023). In LLM Evaluation (Chiang and Lee 2023), they also conducted the experiments using two tasks. Considering that our paper is a short paper and the two prior works are long paper, the number of results/tasks are comparable to the two prior works.
>
> > More targeted conclusions are missing, e.g. recommendations for further prompting.
>
> - In Section 3.3 (title: empirical guidelines), we provide concrete guidelines on recommendations for users. We restate the conclusions here: **Always ask ChatGPT to explain/analyze when rating; Auto-CoT is optional**. The other two reviewers acknowledge the contribution of these guidelines.
>
> Answers to questions:
> ===
> >  I would expand the limitations section on potential misuses (perhaps in conjuction with the ethical analysis)
>
> - We will discuss the potential misuses, including the possibility that human raters on Mturk or Prolific might use ChatGPT for ratings. We will also discuss some concerns about ChatGPT making human raters lossing their jobs.
>
> > How robust you think the results are given the analysis? What are your strongest conclusions?
>
> - The results are robust for the tasks we tested. In the response to Reviewer Efov, we also say that our results are robust to the number of responses sampled from the LLM, the temperature used during generation, and the prompts given to the LLM. As a responsible scientific paper, we cannot overclaim that the observation will be general to the tasks we did not test.
> Our paper's strongest and most useful conclusion is that “asking the LLM to explain its ratings consistently improves the correlation between the ChatGPT and human ratings and pushes state-of-the-art (SoTA) correlations on two meta-evaluation datasets.”
>
> Summary
> ===
> We believe the above responses address all the concerns of robustness and reproducibility. We also restate the very useful takeaway in the paper to illustrate the value of our paper. We hope the reviewer can kindly raise the soundness, excitement, and reproducibility score based on responses.

---

### Meta-Review · Area_Chair_jZcR · 2023-09-17

**Recommendation:** 2

**Metareview:**

This paper explores the task of using LLMs to evaluate text quality. In particular, they study how components of LLM evaluation and G-Eval contribute to correlations with human raters. They have some interesting findings, e.g., that asking ChatGPT to explain it's rating improves correlations with human raters.
One main concern about this paper is the use of only ChatGPT. While the authors state why they made this decision in their rebuttal, the framing (especially early in the paper, e.g., the abstract) leads one to believe that their findings are more generalizable. The authors present some new results in the rebuttal, but there is missing information making these results seem rushed and incomplete (e.g., lack of a p-value when stating that something is statistically significant). I am concerned about the amount of content in the paper as well, which was mentioned by multiple reviewers. The reader shouldn't have to flip between the paper and the abstract to find details that are crucial to understanding the paper, and the fact that the authors include a note in the limitations section stating "We cannot include all the experiment results in the main content, and we need to make our language as concise as possible due to the page limit" is telling. The authors have made a lot of promises about what they can add to the final page for a camera-ready, but they have also included a huge amount of new results in their rebuttals, and space seems like it will continue to be an issue.

---

### Decision · Program_Chairs · 2023-10-07

**Decision:**

Accept-Findings

**Comment:**

This paper explores the task of using LLMs to evaluate text quality. In particular, they study how components of LLM evaluation and G-Eval contribute to correlations with human raters. They have some interesting findings, e.g., that asking ChatGPT to explain it's rating improves correlations with human raters.
One main concern about this paper is the use of only ChatGPT. While the authors state why they made this decision in their rebuttal, the framing (especially early in the paper, e.g., the abstract) leads one to believe that their findings are more generalizable. The authors present some new results in the rebuttal, but there is missing information making these results seem rushed and incomplete (e.g., lack of a p-value when stating that something is statistically significant). I am concerned about the amount of content in the paper as well, which was mentioned by multiple reviewers. The reader shouldn't have to flip between the paper and the abstract to find details that are crucial to understanding the paper, and the fact that the authors include a note in the limitations section stating "We cannot include all the experiment results in the main content, and we need to make our language as concise as possible due to the page limit" is telling. The authors have made a lot of promises about what they can add to the final page for a camera-ready, but they have also included a huge amount of new results in their rebuttals, and space seems like it will continue to be an issue.